Vertical distribution characteristics of soil organic carbon and vegetation types under different elevation gradients in Cangshan, Dali

Yang Xue 1 2
Xu Jianhong 1 2
Wang Huifang 1 2
Quan Hong 1 2
Yu Huijuan 1 2
Luan Junda 1 2
Wang Dishan 1 2
Li Yuancheng 1 2 3
Lv Dongpeng lvdp1010@163.com 1 2
1 College of Agronomy and Biological Sciences, Dali University , Dali, Yunnan , China
2 Key Laboratory of Ecological Microbial Remediation Technology of Yunnan Higher Education Institutes, Dali University , Dali, Yunnan , China
3 Faculty of Metallurgical and Energy Engineering, Kunming University of Science and Technology , Kunming, Yunnan , China
Shahzad Tanvir
Electronic publication date: 2024 Jan 4
Publication date: 2024
Volume: 12
Electronic Location ID: e16686
Received 2023 Aug 9; Accepted 2023 Nov 26
Copyright: ©2024 Yang et al.
Copyright year: 2024
Copyright holder: Yang et al.
License: This is an open access article distributed under the terms of the Creative Commons Attribution License, which permits unrestricted use, distribution, reproduction and adaptation in any medium and for any purpose provided that it is properly attributed. For attribution, the original author(s), title, publication source (PeerJ) and either DOI or URL of the article must be cited.
License URL: https://creativecommons.org/licenses/by/4.0/

Keywords: SOC, SOC components, Vertical distribution, Vegetation types

Funding: Research Project on Ecosystem Service Function of Tailwater Wetland in the Three River Merging Area KY1916104440 This work was supported by the Research Project on Ecosystem Service Function of Tailwater Wetland in the Three River Merging Area (KY1916104440). The funders had no role in study design, data collection and analysis, decision to publish, or preparation of the manuscript.

==============================
Background

The Cangshan National Nature Reserve of Dali City was adopted as the research object to clarify the vertical distribution characteristics of soil organic carbon (SOC) and vegetation types at different elevations in western Yunnan.

Methods

The contents of SOC, light fraction organic carbon (LFOC), heavy fraction organic carbon (HFOC), and water-soluble organic carbon (WSOC) in the 0–30 cm soil layer at different elevations (2,400, 2,600, 2,800, 3,000, 3,200, 3,400, and 3,600 m) were determined, and the above-ground vegetation types at different elevations were investigated.

Results

Results showed that the SOC content was the highest in 0–20 cm surface soil and gradually decreased with the deepening of the soil layer. It increased then decreased with the increase in elevation, and it peaked at 3,000 m. The LFOC content was between 1.28 and 7.3515 g kg−1. It exhibited a decreasing trend and little change in profile distribution. The HFOC content ranged between 12.9727 and 23.3708 g kg−1; it increased then decreased with the increase in profile depth. The WSOC content was between 235.5783 and 392.3925 mg kg−1, and the response sensitivity to elevation change was weak. With the increase in elevation, WSOC/SOC and LFOC/SOC showed a similar trend, whereas HFOC presented an opposite trend. This observation indicates that the active organic carbon content at 3,600 m was lower than that at 2,400 m, and the middle elevation was conducive to the storage of active organic carbon. Meanwhile, the physical and chemical properties of soil affected the distribution of organic carbon to a certain extent. The vegetation type survey showed that the above-ground dominant species within 2,400–2,800 m were Pinus yunnanensis and Pinus armandii. Many evergreen and mixed coniferous broadleaf forests were distributed from 3,000 m to 3,200 m. Species of Abies delavayi were mainly distributed from 3,400 m to 3,600 m. This research serves as a reference for the study of forest soil carbon stability in high-elevation areas and plays an important role in formulating reasonable land use management policies, protecting forest soil, reducing organic carbon loss, and investigating the carbon sequestration stability of forest ecosystems.

Introduction

Soil carbon pool is the largest carbon pool in the terrestrial ecosystem and the best options for carbon storage in terrestrial ecosystems, which plays a crucial role in the global carbon cycle (Liu et al., 2022; Zhang et al., 2023a). In recent years, carbon storage and its influencing factors have been the focus of attention. Soil organic carbon (SOC), as a key indicator of soil quality, is crucial for maintaining soil fertility, improving the soil structure, increasing vegetation coverage, and mitigating global warming enhancing soil’s water holding capacity, and improving soil productivity (Wang et al., 2021; Alekseev & Abakumov, 2022; Chen et al., 2022). Forest soils are one of the important organic carbon pools in terrestrial ecosystems. Minor changes in SOC stocks can substantially affect the entire terrestrial ecosystem and atmospheric carbon dioxide concentrations (Zhou et al., 2021; Song et al., 2022). Generally, SOC is composed of different carbon components, which can be divided into: active and inert organic carbon, according to their stability differences in soil (Liu et al., 2021). Light fraction organic carbon (LFOC) and water-soluble organic carbon (WSOC) are important indicators to characterize active organic carbon, and heavy fraction organic carbon (HFOC) is the main part of inert organic carbon, with stable and complex structure and strong anti-interference ability, the changes of SOC content and components are important indicators for studying the evolution of soil quality, and jointly regulate the stability of the forest soil carbon pool. In recent years, carbon components have become a hot spot in soil carbon pool research (Zhao et al., 2018; Li et al., 2017). Thus, the spatial distribution characteristics of SOC and its component content need to be accurately assessed to enhance soil carbon sequestration and mitigate global environmental changes (Bai & Zhou, 2020), and evaluating and predicting the stability mechanism of organic carbon components.

The spatial distribution of SOC is affected by climate change (Yang et al., 2016), human activity, plant type (Gray, Bishop & Wilson, 2015), temperature, precipitation and terrain (Zhu et al., 2017; Wang et al., 2019; Zhang et al., 2019a; Ge et al., 2020), and other factors (Toh et al., 2020; Xue et al., 2019; Conforti et al., 2016). Climate change may drive soil–carbon dynamics. Permafrost thawing due to warming causes previously stored organic carbon to be decomposed by microorganisms and eventually released into the atmosphere in the form of CO2 (Li et al., 2020; Perez-Mon et al., 2022; Chang et al., 2021). Warming increases the altitudinal range of thermophilic species (Steinbauer et al., 2018), accelerates the retreat of alpine glaciers (Hohensinner et al., 2021), and alters primary productivity (Lu et al., 2021). Human activities (Carboni et al., 2018; Urbina et al., 2020; Ameztegui et al., 2021), afforestation (Ortiz et al., 2016; Hong et al., 2020; Piao et al., 2020; Bastin et al., 2019), deforestation of large areas of virgin forests (Santini et al., 2020), and grazing (Zhang et al., 2020; Goenster-Jordan et al., 2021; Abdalla et al., 2018; Wang et al., 2022; Gao et al., 2018), have also altered the amount of stored organic carbon. Differences in elevation and hydrothermal combinations lead to remarkable differences in plant communities (Zhang et al., 2021), which in turn lead to substantial spatial heterogeneity in SOC stocks (Zhang et al., 2023b). Previous studies on organic matter and its components mostly focused on the influence of different land cover and land use modes on the stability of SOC. Dinakaran et al. (2018) studied the stability of SOC under four different land covers (pasture land, mixed cover, chirpine and agricultural land) and concluded that the maximum SOC stock was observed under a mixed land cover. Wang et al. (2018) studied the distribution of SOC in the low hilly land in the southern part, sandy land in the middle part and an alluvial plain in the northern part, and concluded that SOC initially decreased and then increased from the north to the south, the highest were generally in the south, with the lowest generally in the middle part. Many scholars have studied the impact of LFOC under different land use modes and found that the LFOC content of forest soil is higher than that of cultivated land, the LFOC content of soil decreases significantly after natural grassland and forest land are transformed into cultivated land, and the LFOC content and its distribution ratio can be significantly increased by returning farmland to forest and grassland (Shang et al., 2014). Most studies on SOC content and distribution characteristics have focused on a single influencing factor and ignored the effects of elevation and vegetation type on organic carbon distribution. Therefore, studying the distribution characteristics of SOC and component content and above-ground vegetation types at different elevations is essential for the accurate estimation of regional SOC storage.

Cangshan is a typical area for studying different vertical zones and vegetation types in mountainous areas and has a unique geographical environment and ecosystem functions. Therefore, layer-by-layer analysis of the spatial distribution characteristics of SOC is essential for the accurate determination of the vertical distribution and influencing factors of organic carbon storage in western Yunnan because it strengthens the research on the storage and distribution characteristics of SOC in this area (Zhang, Zhu & Shao, 2019b). The specific contributions of this study are as follows: (1) the distribution characteristics of SOC content along different elevations and soil profiles in the study area were evaluated, (2) the distribution of SOC components (HFOC, LFOC and WSOC) in different elevations and soil profiles was determined, and (3) the above-ground vegetation types of SOC and its component contents under different elevation gradients were investigated.

Materials & Methods

Study area

The Cangshan study area (25°34′–26°00′N, 99°55′–100°12′E) is located in Dali Bai Autonomous Prefecture, western Yunnan Province, across Dali City. It has a subtropical southwest monsoon climate. The average annual temperature is 15.5 °C, and the annual precipitation is more than 1,000 mm. Cangshan has a steep terrain that consists of 19 peaks. The highest point, Malone Peak, is 4,122 m above sea level. In this study, a spline with a length of about 1,200 m near Cangshan National Geological Park in Dali was selected. The elevation range was 2,400–3,600 m. Sampling sites were set every 200 m along the transect, and three parallel sampling sites within a range of 10 km2 were selected for soil sample collection (Fig. 1).

Figure 1 Location of the study area in Dali and distribution of sampling points.

A spline with a length of about 1,200 m near Cangshan National Geological Park in Dali was selected, the elevation range was 2,400–3,600 m. The data is provided by Geospatial Data Cloud site, Computer Network Information Center, Chinese Academy of Sciences (http://www.gscloud.cn).

Collection and pretreatment of soil samples

On the basis of numerous field investigations, soil samples were collected from 2,400, 2,600, 2,800, 3,000, 3,200, 3,400, and 3,600 m elevations by using the hand-held stainless steel drills bit from December 2022 to February 2023. The sampling point information is shown in Table 1. Three parallel sample points were selected from each sampling elevation region. The soil samples were collected from a depth of 0–30 cm. The litter on the soil surface was removed before sampling, and soil samples were collected successively at sampling depths of 0–5, 5–10, 10–20, and 20–30 cm. The vegetation types were recorded during soil sample collection. The soil profile was divided into layers 0–5, 5–10, 10–20, and 20–30 cm on the basis of sampling depth. The soil samples were packed into sample bags, numbered, and brought back to the laboratory for subsequent analysis. Moreover, GPS (G120BD, Guangdong) was used to determine the elevation (Alt), latitude (Lat), and longitude (Lon), and soil water content (SWC) was measured with a soil moisture detector (SOONDA TR-6, Guangdong). The soil samples’ physicochemical properties are shown in Table 2.

Table 1 Characteristics and soil physicochemical properties of the study sites at different elevations along the slope of Cangshan, China.

Theoretical/ actual elevation (m)	Level	GPS location	Vegetation types	Soil types	
2,400/2,433	High	25°41′36.46″N,100°7′43.47″E	Pinus yunnanensis.	red soil	
2,600/2,665	High	25°41′23.04″N,100°7′23.35″E	Mixed forest (Pinus yunnanensis. and Pinus armandii.)	yellow brown soil	
2,800/2,814	High	25°41′15″N,100°7′9.55″E	Pinus armandii.	yellow brown soil	
3,000/3,076	High	25°41′5.73″N,100°6′45.13″E	Evergreen Broadleaf Forest (Lithocarpus leucostachyus A. Camus + Schima argentea Pritz. ex Diels)	brown soil	
3,200/3,191	High	25°41′3.55″N,100°6′37.84″E	Mixed Coniferous Broad Leaved Forest (Tsuga dumosa (D. Don) Eichler + Betula delavayi Franch.)	dark brown soil	
3,400/3,478	High	25°40′55″N,100°6′19.39″E	Abies delavayi- Rhododendron simsii Planch.	dark brown soil	
3,600/3,652	High	25°40′48.15″N,100°5′57.24″E	Abies delavayi.	dark brown soil	

Table 2 Soil physicochemical properties along a range of elevations in Cangshan.

Theoretical/ actual elevation
(m)	Soil depth
(cm)	pH	EC
(us cm−1)	Soil water content (%)	Soil organic matter (%)	
2,400/2,433	0–5	4.38 ± 0.207a	140 ± 12.124a	22.38 ± 7.932a	19.30 ± 4.017a	
5–10	4.78 ± 0.015b	117 ± 1.000b	19.09 ± 5.472ab	15.36 ± 0.971a	
10–20	5.05 ± 0.075c	102.67 ± 5.774c	18.14 ± 6.475a	12.55 ± 1.878a	
20–30	5.14 ± 0.053c	96.67 ± 2.082c	18.24 ± 6.927a	11.72 ± 1.516a	
2,600/2,665	0–5	4.71 ± 0.397a	121 ± 22.913a	32.83 ± 9.334ab	27.51 ± 6.264b	
5–10	5.02 ± 0.110ab	102.33 ± 6.506ab	28.77 ± 5.243bc	19.43 ± 2.454ab	
10–20	5.21 ± 0.059b	91.67 ± 3.786b	27.96 ± 3.866abc	16.80 ± 2.341a	
20–30	5.30 ± 0.246b	88.67 ± 15.503b	28.15 ± 3.384abc	16.11 ± 1.463a	
2,800/2,814	0–5	5.64 ± 0.501a	79.67 ± 7.371a	43.91 ± 2.234b	37.24 ± 8.259c	
5–10	5.55 ± 0.131a	71.67 ± 7.638a	33.68 ± 0.819cd	23.03 ± 0.617b	
10–20	5.63 ± 0.397a	79.67 ± 10.408a	32.46 ± 2.152cd	18.59 ± 2.164a	
20–30	5.52 ± 0.108a	74.67 ± 6.658a	32.47 ± 2.838bc	18.26 ± 2.007a	
3,000/3,076	0–5	6.60 ± 0.299a	12.67 ± 24.194a	24.08 ± 13.506ab	16.36 ± 0.878a	
5–10	6.79 ± 0.123a	1.33 ± 9.815a	22.15 ± 6.570ab	13.92 ± 1.593a	
10–20	6.65 ± 0.120a	13.00 ± 4.359a	22.57 ± 7.732abc	13.74 ± 0.919a	
20–30	6.72 ± 0.035a	5.67 ± 5.132a	21.33 ± 14.626ab	12.57 ± 1.639a	
3,200/3,191	0–5	5.92 ± 0.237a	50.33 ± 13.577a	30.36 ± 2.245ab	15.14 ± 0.996a	
5–10	5.55 ± 0.229ab	71.00 ± 13.115ab	28.59 ± 7.157bc	16.69 ± 1.548ab	
10–20	5.58 ± 0.306ab	70.00 ± 16.371ab	29.17 ± 5.463bc	17.06 ± 2.905a	
20–30	5.40 ± 0.254c	80.33 ± 14.434b	27.62 ± 5.911abc	16.72 ± 5.847a	
3,400/3,478	0–5	4.88 ± 0.124a	111.33 ± 6.351a	15.05 ± 2.939a	18.08 ± 1.630a	
5–10	4.81 ± 0.105a	115.33 ± 6.028a	16.31 ± 3.079a	16.21 ± 4.874ab	
10–20	4.95 ± 0.164ab	107.33 ± 9.452ab	19.44 ± 3.664ab	16.37 ± 4.743a	
20–30	5.12 ± 0.035b	98.00 ± 3.606b	21.90 ± 4.218ab	16.19 ± 4.430a	
3,600/3,652	0–5	4.37 ± 0.316a	140.67 ± 17.388a	34.93 ± 20.521ab	18.25 ± 4.145a	
5–10	4.39 ± 0.165a	139.33 ± 9.074a	39.15 ± 6.621d	16.07 ± 7.479ab	
10–20	4.44 ± 0.173a	136.67 ± 9.815a	39.75 ± 7.547d	13.38 ± 6.163a	
20–30	4.54 ± 0.08a	131.00 ± 5.000a	39.15 ± 5.220c	12.10 ± 4.181a	

Measurement and analysis of soil samples

The soil samples were air dried, sieved through a two mm sieve (10 mesh), and weighed for the determination of indicators.

SOC content

SOC content was determined through potassium dichromate titration with reference to the “Potassium Dichromate Oxidation Spectrophotometric Method” (HJ 615-2011) (Bierer et al., 2021; Bahadori & Tofighi, 2016; Weaver, Birdsey & Lugo, 1987; Beltrame et al., 2016; Lã et al., 2023; Tatzber et al., 2015; Mustapha et al., 2023), and light fraction organic carbon (LFOC) and heavy fraction organic carbon (HFOC) were extracted with the relative density method (Azam et al., 2020). Water-soluble organic carbon (WSOC) was extracted at a soil–water ratio of 1:6, and a TOC analyzer (TOC-L; Shimadzu, Tokyo, Japan) was used.

Pretreatment of soil samples: Twenty grams of the air-dried soil samples that had passed through a two mm sieve were placed in a 250 mL plastic bottle with a fine mouth, and 50 mL of sodium iodide solution with a density of 1.8 g cm−3 was added. The samples were shaken at 200 rpm for 2 min on a vibrating machine. Substances that adhered to the wall of the tube were washed with an additional 10 mL sodium iodide solution. The tube was left overnight then centrifuged at 5,000 rpm for 30 min. Immediately after centrifugation, the supernatant was poured into the filter device of a 0.45 um filter membrane for vacuum filtration, washed with 50 mL of 0.5M CaCl2 solution, and rinsed with 50 mL of distilled water. The 0.45 um component was dried and weighed for 24 h at 50°C. The organic carbon of the dried sample was analyzed with the Black–Wely method after grinding.

Soil recombination and light group separation: Exactly 100 g of the air-dried soil samples were weighed and divided into three equal parts, which were placed in heavy liquid with a density of 1.70 g cm−3, shaken by hand for 5 min, shaken ultrasonically at 400 Jml−1 for 3 min, centrifuged, siphoned to obtain the supernatant, and filtered; this process was repeated three times. The obtained samples were washed with 100 mL 0.01 mol L−1 of CaCl2 solution and then repeatedly with 200 mL of distilled water to obtain the light group. The rest was reconstituted, washed with 100 mL 0.01 mol L−1 of CaCl2 solution, and rinsed repeatedly with 200 mL of distilled water. The sample recovery rate was above 95%.

Potassium dichromate external heating of organic carbon: Exactly 2 g of the air-dried soil sample was weighed, subjected to a 0.25 mm sieve, placed in a 250 mL triangular bottle, added with 10 mL of 1N potassium dichromate, mixed evenly, added with 20 mL of concentrated sulfuric acid, boiled for 1 min by “blowing bubbles” steam, cooled for half an hour, added with 50 mL of distilled water, added with three drops of linfenolin indicator, and titrated with 0.5 N ferrous sulfate solution. FeSO4 was added drop by drop until the color became gray–blue–green. Two blank measurements were made at the same time (no added soil samples).

(1) SOC%=V0−V×N×0.003×1.3m×100%.

(2) SOCgkg−1=V0−V×N×0.003×1.3m×1000.

where V0 is the volume of ferrous sulfate consumed by two blanks, V is the volume of ferrous sulfate consumed by the sample, m is the drying soil weight, N is the equivalent concentration of FeSO4, and 0.003 is the number of grams of 1 mg equivalent C.

Soil pH, water, and organic matter

Soil pH: Determination of soil pH was performed with reference to the “Determination of Soil pH Potentiometric Method” (HJ 962-2018). Ten grams of the air-dried soil samples were subjected to a 20-mesh sieve and placed in a 50 mL beaker, which was then added with 25 mL of water (soil–water ratio of 1:2.5). The container was sealed with a sealing film, vibrated violently in a horizontal oscillator for 5 min, and allowed to stand for 30 min. The pH of the supernatant was determined with a pH meter within 1 h.

Soil water content (SWC): Soil samples were collected successively at sampling depths of 0–5, 5–10, 10–20, and 20–30 cm by using a soil drill, and the wet soil weight onsite was measured. The samples were brought back to the laboratory for drying at 105 °C, the dry weight was measured, and the weight of the box was determined to calculate SWC. The formula is as follows: (3) SWC=W−DW−B×100%

where W is the wet soil weight, D is the dry weight, B is the weight of the box, the units are all g.

Soil organic matter (SOM): SOM content was determined with the “Solid Waste–Determination of Organic Matter and Ignition Loss Method” (HJ 761-2015). The crucible was burned to a constant weight in a muffle furnace at 600°C in advance, and 2 g of air-dried, ground, and sifted soil samples were weighed in the crucible and burned in the muffle furnace at 600 °C for 3 h. After cooling, the samples were removed from the furnace and weighed. SOM content was calculated by measuring the change in the weight of the soil before and after burning. The SOM calculation formula is as follows: (4) SOM=m2−m3m1×100%

where the weight of the crucible was recorded as m0, the weight of the soil was recorded as m1, the weight of the soil and crucible was recorded as m2, and the weight of the soil and crucible after burning was recorded as m3, the units are all g.

Vegetation types

Above-ground vegetation types were recorded when the soil samples were collected. These vegetation types were used to identify the vertical distribution types and spatial changes in vegetation at different elevations.

Statistical analysis

One-way analysis of variance (ANOVA) followed by least significant difference (LSD) tests were used to analyze the significant difference, taking p-values lower than 0.05 as the significance level, and all representations of variance are standard errors (±SE). The figures were generated using Origin85 (Origin Lab), the topographic map of sampling points was drawn by Arcgis. All analyses were conducted using SPSS 26 (IBM, Armonk, NY).

Results

Changes in SOC content at different elevations

The SOC content in Cangshan showed regional differences in distribution characteristics. As indicated in Fig. 2, in the study area, the organic carbon content at each sampling point generally decreased with the increase in soil depth. At the same sampling point, the SOC content of surface soil (0–20 cm) was the highest, and the SOC content in layer A+B+C+D varied between 80.7625 and 96.51525 g kg−1. The average SOC at the different sampling points was in the order of 3,600 (80.7625) <3,400 (86.71) <3,200 (88.127) <2,400 (90.662) <2,600 (90.85583333) <2,800 (96.24875) <3,000 (96.51525). The SOC distribution at the different elevations had similar and different points. The SOC content was the highest in layer A+B+C (0–20 cm). The SOC content increased then decreased from layers A to D. At different elevations (2,400–3,600 m), the SOC content varied with the increase in soil depth. A gradually increasing trend was observed from layers A to B, which increased by 19.20%, 13.05%, 21.32%, 21.46%, −1.73%, 4.09%, and 19.95%. Layers B and C decreased by 29.37%, −1.44%, 8.83%, 18.45%, −14.62%, 4.71%, and 0.97%, indicating a gradually decreasing trend. Layers C and D decreased by 13.54%, 29.07%, 24.49%, −2.96%, 1.70%, 27.66%, and 18.62%, showing a sharp decline. The difference between different soil layers below 2,600 m is significant, indicating that soil profile depth has a significant impact on SOC, while the difference between different soil layers above 2,800 m is basically not significant, indicating that soil profile depth has no significant impact on SOC, and SOC content above 2,800 m is mainly affected by other environmental factors.

Figure 2 Vertical profile changes in SOC content at different sampling points.

(A) to (G) present the distribution maps of SOC at different elevations. (H) presents the distribution of SOC at different sampling points in the 3D dimension. Different letters indicate differences between different layers at the same elevation.

Vertical profile changes in SOC components at different elevations

Figure 3 shows that the distribution of SOC components in the soil profiles varied at different elevations. Figure 3A indicates that the LFOC content ranged from 1.287 g kg−1 to 7.3515 g kg−1. The average LFOC content at 2,400, 2,600, 2,800, 3,000, 3,200, 3,400, and 3,600 m above sea level was 4.2283, 4.4476, 4.5435, 4.6134, 5.1788, 4.6393, and 3.6156 g kg−1, respectively. The LFOC content generally showed a decreasing trend, and the LFOC profile distribution did not change much. Figure 3B shows that the HFOC content ranged from 12.9727 g kg−1 to 23.3708 g kg−1. The average HFOC content at 2,400, 2,600, 2,800, 3,000, 3,200, 3,400, and 3,600 m were 18.4373, 18.2663, 19.5853, 19.4488, 16.8529, 17.0382, and 16.5750 g kg−1, respectively. The HFOC profile distribution at different elevations was similar to the SOC distribution; it initially increased and then decreased with the increase in profile depth, indicating that the different elevations played an important role in the turnover process of SOC. Figure 3C shows that the WSOC content ranged from 235.5783 mg kg−1 to 392.3925 mg kg−1 at elevations of 2,400, 2,600, 2,800, 3,000, 3,200, 3,400, and 3,600, and the average WSOC content was 295.0617, 338.3092, 392.3925, 309.2242, 270.0225, 345.96, and 235.5783 mg kg−1, respectively. Different soil layers of LFOC showed different significant differences under different elevation gradients. There were significant differences between different elevations under 0–5 cm soil profile, no significant differences between 5–10 and 10–20 cm soil profiles, and significant differences between 20–30 cm soil profiles. The results showed that LFOC accumulation was positive in 0–5 and 20–30 cm soil profiles, HFOC accumulation was positive in 5–10 and 20–30 cm soil profiles, and WSOC accumulation was positive in 0–5, 10–20 and 20–30 cm soil profiles, and there was no significant difference in other layers, which may be related to the amount of plant residues and surface vegetation litter.

Figure 3 Distribution characteristics of SOC components at different elevations.

(A) Distribution of LFOC content, (B) distribution of HFOC content, and (C) distribution of WSOC content. Different letters indicate the variation of the same soil layer at different elevations.

Distribution ratio of organic carbon components in soil profiles at different elevations

Distribution ratio of SOC components at different elevations are shown in Table 3. The LFOC/SOC values at the different elevations ranged from 13.53% to 24.09%, the HFOC/SOC values ranged from 74.41% to 86.47%, and the WSOC/SOC values ranged from 0.97% to 6.02%. HFOC was the main component of SOC and accounted for the largest proportion. No significant difference in LFOC/SOC, HFOC/SOC, and WSOC/SOC was observed under different elevation gradients of 5–10, 10–20, and 20–30 cm, but the LFOC/SOC and HFOC/SOC in the 0–5 cm soil layer differed. With the increase in elevation, WSOC/SOC and LFOC/SOC showed a similar change trend in general, but the change trend of HFOC was the opposite. These results indicate that the active organic carbon content at 3,600 m exhibited a lower trend than that at 2,400 m, and the storage of active organic carbon content at middle elevation was favorable.

Table 3 Distribution ratio of SOC components at different elevations.

Elevation
(m)	Soil depth
(cm)	LFOC/SOC
(%)	HFOC/SOC
(%)	WSOC/SOC
(%)	
2,400	0–5	16.38ab	83.62ab	2.69a	
5–10	19.33a	80.67a	2.08a	
10–20	22.77a	77.23a	1.07a	
20–30	14.76a	85.24a	1.15a	
2,600	0–5	22.17ab	77.83ab	2.90a	
5–10	21.67a	78.33a	1.94a	
10–20	15.88a	84.12a	1.52a	
20–30	18.49a	81.51a	2.11a	
2,800	0–5	19.33ab	80.67ab	1.81a	
5–10	18.95a	81.05a	1.74a	
10–20	16.81a	83.19a	1.82a	
20–30	19.32a	80.68a	1.74a	
3,000	0–5	19.04ab	80.96ab	1.51a	
5–10	17.26a	82.74a	2.00a	
10–20	20.30a	79.70a	1.09a	
20–30	20.58a	79.42a	0.97a	
3,200	0–5	25.59b	74.41a	2.28a	
5–10	22.33a	77.67a	2.15a	
10–20	22.15a	77.85a	1.59a	
20–30	24.09a	75.91a	1.35a	
3,400	0–5	23.77ab	76.23ab	4.28a	
5–10	16.91a	83.09a	4.99a	
10–20	23.34a	76.66a	3.80a	
20–30	21.59a	78.41a	6.02a	
3,600	0–5	14.92a	85.08b	1.71a	
5–10	19.31a	80.69a	1.50a	
10–20	22.50a	77.50a	1.10a	
20–30	13.53a	86.47a	1.83a	

Factors that affect SOC content and its components

Interaction of SOC content and soil physicochemical properties at different elevations

The content distribution of SOC and its components under different soil physical and chemical properties at different elevations in the study area is shown in Fig. 4. The content distribution of SOC and its components under different soil physical and chemical properties (pH, SWC, and SOM) differed. The effects between pH and LFOC, SWC and HFOC, SOM and WSOC were obvious.

Figure 4 Interaction of SOC content and soil physicochemical properties at different elevations.

(A) to (D) describe how pH interacts with SOC and its component content, (E) to (H) describe how SOM interacts with SOC and its component content, and (I) to (L) describe how SWC interacts with SOC and its component content.

Relationship between SOC and soil properties

Figure 5 presents the relationship between SOC content and soil basic physical and chemical properties. In terms of soil basic physical and chemical properties, pH, EC, and SWC had no significant correlation with SOC content, but SOM and WSOC had a significant positive correlation (p < 0.001). The content of each organic carbon component was correlated with SOC content. Specifically, SOC and LFOC had a significant positive correlation (p < 0.001), HFOC also had a significant positive correlation (p=0.01), and SOC was correlated with WSOC (p < 0.05). The organic carbon components also exhibited different degrees of correlation. LFOC and HFOC were significantly correlated (p = 0.01), and HFOC was correlated with WSOC (p < 0.05).

Figure 5 Relationship among SOC, LFOC, HFOC, and WSOC contents and the influencing factors.

*** means significant at p < 0.001, ** means significant at p < 0.01, and * means significant at p < 0.05. PH, soil pH; SWC, soil water content; SOM, soil organic matter.

Distribution types of above-ground vegetation at different elevations

The distribution of above-ground vegetation at different elevations in the study area is shown in Fig. 6. The ground vegetation at 2,400–2,800 m was mainly pine forest, in which Pinus yunnanensis and Pinus armandii were the dominant species. Many evergreen and mixed coniferous broadleaved forests were distributed at 3,000–3,200 m. The common plant species were Lithocarpus leucostachyus A. Camus + Schima argentea Pritz. ex Diels and Tsuga dumosa (D. Don) Eichler + Betula delavayi Franch. With the increase in elevation, Abies delavayi became the main forest at 3,400–3,600 m, and Abies delavayi-Rhododendron simsii Planch was the main common plant species. The vegetation distribution along the sampling sites mainly showed that Pinus yunnanensis was widely distributed below 2,500 m. Most of the species were positive and barren tolerant, and there were patches of shrub grass slopes in some areas. The vegetation at 2,500–2,900 m is dominated by Pinus yunnanensis and Pinus armandii. The mountains in this area are steep and there is a small amount of evergreen broad-leaved forest, indicating that this is a transitional zone from Pinus yunnanensis and Pinus armandii forest to evergreen broad-leaved forest. From 2,900 to 3,400 m, it is mainly mixed forest, and many areas are replaced by Abies delavayi., but still maintain the mixed forest phase, and then transition to Cangshan Abies delavayi Diels- Rhododendron simsii Planch. forest at higher elevations. 3,400–3,700 m are mainly Abies delavayi Diels-Rhododendron simsii Planch forest, which has less environmental damage and similar community composition to other locations in Cangshan.

Figure 6 Distribution characteristics of above-ground vegetation at different elevations.

The main distribution patterns of vegetation are as follows: (1) The native vegetation below 2,900 m is more affected by human disturbance, and the environmental damage degree is greater than that in the high elevation area, which is as follows: the evergreen broad-leaved forest is replaced by Pinus yunnanensis. Most of the wet evergreen broad-leaved forest was replaced by the warm coniferous Pinus armandii, and there were a few evergreen broad-leaved forests in the transition zone, (2) The dominant vegetation of 2,900–3,400 m is Lithocarpus leucostachyus A. Camus, Schima argentea Pritz. ex Diels, Tsuga dumosa (D. Don) Eichler, Betula delavayi Franch and Rhododendron simsii Planch above Abies delavayi, and the native vegetation in this area is Picea asperata Mast forest. Due to the distribution elevation of Picea asperata Mast. is slightly lower than that of Abies delavayi. and its wood is better, the Picea asperata Mast. forest has largely disappeared under large-scale human disturbance, and has been replaced by the less disturbed Abies delavayi- Rhododendron simsii Planch. The transition area is usually Cangshan Abies delavayi, which basically preserves the main species of native vegetation - Tsuga dumosa (D. Don) Eichler, Lithocarpus leucostachyus A. Camus and Abies delavayi and (3) In the zone above 3,400 m, man-made damage is less, the characteristics of native vegetation are more obvious, and the vegetation characteristics formed by long-term natural evolution are more retained, and the Abies delavayi. is limited by 3,400 m. The SOC content in the study area was affected by the type of above-ground vegetation. In general, the level of human activity in the 2,400–3,600 m range was low, the above-ground vegetation productivity was high, and the decomposition of soil litter was slow, which was conducive to the accumulation of organic matter in the soil.

Discussion

Effects of different elevations on SOC profile distribution

In the forest ecosystem, the change of altitude is one of the important factors affecting SOC (Chen et al., 2016). The change of altitude causes the change of vegetation type. With the increase of elevation, the air temperature shows a downward trend, and the lower soil temperature at high elevation results in the lower decomposition rate of SOC (Weaver, Birdsey & Lugo, 1987). Since elevation difference reflects different air temperature and affects vegetation types, which in turn affects SOC content, the impact of elevation difference on SOC content is essentially the influence of climate factors. However, SOC in this study area did not show a trend of gradual increase with the increase of elevation. It showed that SOC content at middle elevation was significantly higher than that at high and low elevations. The main reason for this phenomenon is that the middle region is rich in vegetation (Lasanta et al., 2020; Sokołowska et al., 2020) and rich in soil water resources. In the elevation area of the study area, the vegetation type is mainly mixed forest, and the surface litter produced is the main source of soil surface organic carbon, and the vegetation roots are the important source of soil deep organic carbon. The influence of vegetation type on SOC content is mainly reflected in two aspects: input mode and decomposition path. Meanwhile, the soil in the low-elevation region is affected by water and soil erosion, which take the organic carbon components in the soil and deposit them. The low temperature and humid environment at high elevations limit the decomposition of soil microorganisms, and organic carbon is preserved in the cryogenic frozen soil.

Effects of different elevations on SOC component profile distribution

The influence of elevation difference on SOC components was found in the whole section, and the overall content was WSOC > HFOC > LFOC. However, the profile distribution of LFOC from a to d does not change much, and the profile distribution of HFOC content is similar to that of organic carbon, showing a general trend of first rising and then decreasing with the increase of profile depth (Xi et al., 2020). This may be because different elevations have an important effect on the turnover of SOC, which is related to the amount of plant residues and vegetation litter on the surface. Due to the abundant vegetation and litter on the ground in the middle elevation area, more plant residues are imported into the soil in the form of organic matter. Meanwhile, WSOC content does not change significantly with elevation. There is a significant difference between WSOC at low elevation and WSOC at middle elevation, but no significant difference between WSOC content and WSOC at high elevation. WSOC is a carbon source that can be directly utilized by soil microorganisms and participates in biogeochemical cycles. This result may be due to the fact that microorganisms can quickly convert WSOC into their own carbon (Xiang et al., 2015). The middle-elevation area with abundant vegetation is conducive to the accumulation of organic carbon, and the soil in the low-elevation area is likely to lose organic carbon components due to abundant human activities, which not only reduces the content of LFOC but also impedes the accumulation of HFOC with high stability. The low temperature and humid environment at high elevations limit the decomposition of soil microorganisms, and organic carbon is preserved in the low-temperature frozen soil. Moreover, the level of human activities and the amounts of animal and plant remains at high elevations are low, so the organic carbon content is lower than that in the central region of the study area.

Influencing factors of SOC and its components

No significant correlation was observed among soil pH, EC, SWC, and SOC content possibly because no direct complex exists among soil pH, EC, SWC, and SOC content (Xi et al., 2020). Moreover, this study only conducted sampling in the wet season and lack sample data from other seasons. Therefore, the effect of soil physical and chemical properties on organic carbon content cannot be fully reflected. The relationship between SOC and its component content at different elevations and soil physical and chemical properties needs to be further studied. Soil physical and chemical properties can affect SOC content and the amount of soil dead leaves returned. In this study, soil pH value did not change much at different elevations, and the overall significant difference was not obvious. Changes in soil pH value would change the microbial activity in soil, and then change the accumulation of SOC by affecting the turnover capacity of SOC. The water content is the highest at 3,600 m, and the SOC content is slightly lower than 3,000 m. This may be because there is less human interference at high elevations and a lot of water is frozen in the soil. However, the distribution of above-ground vegetation and litter determine that the SOC content at 3,600 m is slightly lower than 3,000 m. However, SOM content mainly shows that the surface content is higher than other soil layers, which is also directly related to surface contact.

Types of above-ground vegetation at different elevations

Soil at different elevations has different types of vegetation cover, and environmental conditions at different elevations are not exactly the same. The vegetation distribution characteristics of the Cangshan study area generally show that the low-elevation area is more affected by human disturbance, and the environmental damage degree is greater than that of the high-elevation area. Most of the original vegetation is replaced by existing vegetation (Pinus yunnanensis. and Pinus armandii.), and the middle elevation area shows a mixed forest, while the high-elevation area has less human disturbance, and more original vegetation types are preserved. Plant types at middle elevations produce more litter of different types, which makes the SOC content at middle elevations slightly higher than that at other elevations. Vegetation varies at different elevations, that is, the vegetation cover in the above-ground part of the soil differs, which changes the biomass and organic carbon of above-ground plants in the soil (Bojko & Kabala, 2017). The influence of above-ground vegetation on SOC depends on plant productivity and litter amount. Generally, the decomposition products of litter from above-ground vegetation are transferred to soil, which increases the organic carbon content in soil (Xiong, Zhou & Zhang, 2020). If a large forest area is lost due to human deforestation, the biomass of above-ground vegetation will decrease, and the carbon cycle of soil will change. Studies have shown that when forests are destroyed and transformed into farmland, SOC is lost at different degrees (Tolimir et al., 2020; Fusaro et al., 2019; Vanacker et al., 2022). Therefore, we need to protect forests and promote the accumulation of organic carbon by afforestation and returning farmland to the forest. Doing so could mitigate global warming (Bastin et al., 2019).

Conclusions

(1) SOC and its component contents in the study area were affected by the change in elevation. They increased first then decreased with the increase in elevation, and the organic carbon contents in all sampling points decreased with the increase in soil depth. At the same sampling point, the SOC content in surface soil (0–20 cm) was the highest.

(2) The distributions of SOC components in the soil profiles at different elevations differed. LFOC content showed a decreasing trend, but the distribution of the LFOC profile did not change much. The profile distribution of HFOC content was consistent with that of SOC; it increased first and then decreased with the increase in profile depth, indicating that the different elevations played an important role in the conversion process of SOC. The response sensitivity of WSOC to elevation change was weak.

(3) The above-ground dominant species at 2,400–2,800 m were Pinus yunnanensis and Pinus armandii. Many evergreen and mixed coniferous broadleaved forests were distributed at 3,000–3,200 m, and Abies delavayi was mainly distributed at 3,400–3,600 m. The decomposition products of above-ground vegetation litter were transferred to the soil, which was conducive to SOC storage. We need to strengthen the awareness on forest protection and stop cutting down trees and clearing mountains to create land.

Under the current pressure of carbon neutrality, the existing research is of great importance for accurately assessing the influence of regional carbon sink and the ebb and flow and balance of soil carbon sink, which helps to strengthen the understanding of the response mechanism of Cangshan SOC cycle to elevation action, and is of great significance for accurately predicting and evaluating the dynamic change of regional SOC under elevation gradient. It provides a reference for the formulation of policies and measures to protect Cangshan region, and provides a reference for local carbon trading.

In future studies, the relative contribution of different environmental factors (litter, roots and secretions, microbial residues and metabolites) and specific biological sources to SOC will be studied based on this research area, so as to reveal the mechanism of plant evolution on SOC cycle, and promote the study on the participation of environmental factors in SOC cycle under the influence of multiple factors. In view of the complexity of soil and environmental factors, the phenomena obtained in the laboratory are fully integrated with field experiments to further enhance the objectivity of the research.

Supplemental Information

Supplemental Information 1 Soil pH data under different elevations and soil layers

The data contains parallel values and average values.

Click here for additional data file.

Supplemental Information 2 Distribution of soil organic matter content under different elevations and soil layers

Click here for additional data file.

Supplemental Information 3 Summary of soil moisture distribution data under different elevations and soil layers

Click here for additional data file.

Supplemental Information 4 Distribution data of soil water-soluble organic carbon at different elevations and soil layers

Click here for additional data file.

Supplemental Information 5 Summary of soil organic carbon component distribution and content data under different elevations and soil layers

Click here for additional data file.

Additional Information and Declarations

Competing Interests

Author Contributions

Data Availability

The authors declare there are no competing interests.

Xue Yang conceived and designed the experiments, performed the experiments, analyzed the data, prepared figures and/or tables, and approved the final draft.

Jianhong Xu performed the experiments, prepared figures and/or tables, and approved the final draft.

Huifang Wang performed the experiments, prepared figures and/or tables, and approved the final draft.

Hong Quan analyzed the data, prepared figures and/or tables, and approved the final draft.

Huijuan Yu analyzed the data, prepared figures and/or tables, authored or reviewed drafts of the article, and approved the final draft.

Junda Luan performed the experiments, authored or reviewed drafts of the article, and approved the final draft.

Dishan Wang performed the experiments, authored or reviewed drafts of the article, and approved the final draft.

Yuancheng Li conceived and designed the experiments, authored or reviewed drafts of the article, and approved the final draft.

Dongpeng Lv conceived and designed the experiments, prepared figures and/or tables, and approved the final draft.

The following information was supplied regarding data availability:

The raw data are available in the Supplemental Files.

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
