# Peer review of "Vertical distribution characteristics of soil organic carbon and vegetation types under different elevation gradients in Cangshan, Dali"

_PeerJ, doi:10.7717/peerj.16686_

## Round 0.1 · original submission · Major Revisions

All the comments of the reviewer should be taken into account. As far the Furthermore, the authors are advised to give the soil types and vegetation characteristics.

As far the reviewer 2 is concerned, the authors should particularly do the following.

They should also cite previous field scale studies that have employed Potassium Dichromate Oxidation Spectrophotometric Method in order to establish the validity of the method because doing CHNS analyses at this stage would not be possible and not needed.

Reviewer 1 ·

Basic reporting

Detailed remarks concerning manuscript:
1. I suggest to include the purpose of the studies together with practical application of the obtained data - results
2. Key words: It is not recommended to use as key words the words or phrases used in the title of the manuscript. Please do needed changes.
3. The direction for the future studies should be proposed
4. “Carbon storage within soil is one of the best options for carbon storage in terrestrial ecosystems (Liu et al., 2022). Soil carbon pool is the largest carbon pool in the terrestrial ecosystem and plays a crucial role in the global carbon cycle (Zhang et al., 2023)” (Lines 71-73). I suggest to combine these two sentences.
5. All tables and figures should be clear for the reader without referring to the text of the manuscript. Please add the explanations, incuding these with statistical analysis, where needed.
6. Please check the citation of the bibliography in the text of the manuscript. See that the reference: ”Bojko O, Kabala C. 2017. Organic carbon pools in mountain soilsóSources of variability 443 and predicted changes in relation to climate and land use changes. Catena, 149: 209- 444 220 DOI: 10.1016/j.catena.2016.09.022.” is cited as (Bojko et al., 2017).
7. ”The SOC distribution at the different elevations had similar and different points. The SOC content was the highest in Layer A+B+C (0-20 cm). The SOC content increased then decreased from Layers A to D. The differences were as follows. At different elevations (2,400-600 m), the SOC content varied with the increase in soil depth”. I suggest to remove the sentence ”The differences were as follows”
8. The „Discussion” section should be modified. Some parts of this section sounds like descroption of the results. See. ”The distribution of SOC in the soil profiles varied at different elevations. The content of LFOC ranged from 1.287 g•kg-1 to 7.3515 g•kg-1. From Layers AñD, LFOC content showed a decreasing trend, and the distribution of the LFOC profile did not change much. HFOC content ranged from 12.9727 g•kg-1 to 23.3708 g•kg-1. The profile distribution of HFOC content at different elevations was similar to that of SOC, and the overall distribution trend of HFOC content increased then decreased with the increase in profile depth.”
Line 324-334. „ …the organic carbon content at each sampling point generally decreased with the increase in soil depth. At the same sampling point, the organic carbon content of surface soil (0-326 20 cm) was the highest, and the SOC content in Layer A+B+C+D varied between 80.7625 and 96.51525 g•kg-1. The distribution of SOC at the different elevations had similarities and differences. SOC content was the highest in Layer A+B+C (0ñ20 cm). It increased then 329 decreased from Layers A to D. The differences were as follows. At different elevations (2,400- 330 3,600 m), the SOC content varied with the increase in soil depth. For the geomorphic units at different elevations, the organic carbon content of each geomorphic unit was relatively stable, and the SOC content generally increased and then decreased from 2,400 m to 3,600 m. The organic carbon content in the middle region was slightly higher than that in the upper and lower parts”

Experimental design

no comment

Validity of the findings

no comments

·

Basic reporting

This study is interesting. However, the introduction section lacks basic information about light SOC fractions and heavy SOC fractions. Their importance?. There are no hypotheses or clear objectives. This section is very weak.

Experimental design

There is no soil type or vegetation characteristic information for the study area. How have authors collected soil samples? Auger or pit method? How reliable is the SOC data coming from the Potassium Dichromate Oxidation Spectrophotometric Method? Have authors done any comparisons with the data coming from CHNS analyzer data?

Validity of the findings

There is no statistical analysis of the data.
The discussion section is very poor. Authors must read a lot of literature with respect to light and heavy SOC fractions.

Additional comments

Also, check the SI units in the text and tables. Figures (3-6) are not of good quality.

---

## Round 0.2 · accepted · Accept

The reviewer is satisfied with the improved version.

·

Basic reporting

no comment

Experimental design

no comment

Validity of the findings

no comment

Additional comments

Authors have substantially improved the manuscript. I recommend this paper for publication in this journal.